# Predicting the effect of CRISPR-Cas9-based epigenome editing

Sanjit Singh Batra[1†], Alan Cabrera[2†], Jeffrey P Spence[3†], Jacob Goell[2], Selvalakshmi S Anand[4], Isaac B Hilton[2,4*], Yun S Song[1,5*]

[1]Computer Science Division, University of California, Berkeley, United States; [2]Department of Bioengineering, Rice University, Houston, United States; [3]Department of Genetics, Stanford University, Stanford, United States; [4]Systems, Synthetic, and Physical Biology Graduate Program, Rice University, Houston, United States; [5]Department of Statistics, University of California, Berkeley, Berkeley, United States

*For correspondence:
isaac.hilton@rice.edu (IBH);
yss@berkeley.edu (YSS)

†These authors contributed equally to this work

Competing interest: The authors declare that no competing interests exist.

## eLife Assessment

This study presents an advance in efforts to use histone post-translational modification (PTM) data to model gene expression and predict epigenetic editing activity. Such models are broadly **useful** to the research community, especially ones that can model and predict epigenetic editing activity, which is novel; additionally, the authors have nicely integrated datasets across cell types into their model. The work is mostly **solid**, but it would be strengthened by performing further comparisons to existing methods that predict gene expression from PTM data and from more comprehensive functional validation of model-predicted epigenome editing outcomes beyond dCas9-p300 based perturbations. This work will be of interest to the epigenetics and computational modeling communities.

**Abstract** Epigenetic regulation orchestrates mammalian transcription, but functional links between them remain elusive. To tackle this problem, we use epigenomic and transcriptomic data from 13 ENCODE cell types to train machine learning models to predict gene expression from histone post-translational modifications (PTMs), achieving transcriptome-wide correlations of ~0.70–0.79 for most cell types. Our models recapitulate known associations between histone PTMs and expression patterns, including predicting that acetylation of histone subunit H3 lysine residue 27 (H3K27ac) near the transcription start site (TSS) significantly increases expression levels. To validate this prediction experimentally and investigate how natural vs. engineered deposition of H3K27ac might differentially affect expression, we apply the synthetic dCas9-p300 histone acetyltransferase system to 8 genes in the HEK293T cell line and to 5 genes in the K562 cell line. Further, to facilitate model building, we perform MNase-seq to map genome-wide nucleosome occupancy levels in HEK293T. We observe that our models perform well in accurately ranking relative fold-changes among genes in response to the dCas9-p300 system; however, their ability to rank fold-changes within individual genes is noticeably diminished compared to predicting expression across cell types from their native epigenetic signatures. Our findings highlight the need for more comprehensive genome-scale epigenome editing datasets, better understanding of the actual modifications made by epigenome editing tools, and improved causal models that transfer better from endogenous cellular measurements to perturbation experiments. Together, these improvements would facilitate the ability to understand and predictably control the dynamic human epigenome with consequences for human health.

## Introduction

All cells within a multicellular organism have the same genetic sequence up to a minuscule number of somatic mutations. Yet, many cell types exist with diverse morphological and functional traits. Epigenetics is an important regulator and driver of this diversity by allowing differences in cellular state and gene expression despite having the same genotype (*Taherian Fard and Ragan, 2019*). Indeed, cells traversing the trajectory from pluripotency through terminal differentiation have essentially the same genotype.

Epigenetic modifications such as post-translational modifications (PTMs) to histone proteins are involved in many vital regulatory processes influencing genomic accessibility, nuclear compartmentalization, and transcription factor binding and recognition (*Reik et al., 2001*; *Kouzarides, 2007*; *Gibney and Nolan, 2010*; *Klemm et al., 2019*; *Hafner and Boettiger, 2023*; *Zhang and Reinberg, 2001*). The Histone Code Hypothesis suggests that combinations of different histone PTMs specify distinct chromatin states, thereby regulating gene expression (*Strahl and Allis, 2000*; *Jenuwein and Allis, 2001*).

The field of epigenome editing has produced new tools for understanding the outcomes of epigenetic perturbations that promise to be useful for therapeutics by enabling fine-tuned control of gene expression (*Matharu and Ahituv, 2020*; *Thakore et al., 2016*; *Goell and Hilton, 2021*; *Stricker et al., 2017*). Currently, small molecule drugs are used to potently interfere with epigenetic regulation of gene expression. For example, Vorinostat inhibits histone deacetylases, thereby impacting the epigenetic landscape (*Estey, 2013*; *Yoon and Eom, 2016*). However, small molecules globally disrupt the epigenome and transcriptome and therefore are not suitable for targeting individual dysregulated genes nor clarifying epigenetic regulatory mechanisms (*Swaminathan et al., 2007*). Meanwhile, numerous tools have been designed to harness catalytically dead Cas9 (dCas9) to target epigenetic modifiers to DNA sequences encoded in guide RNAs (gRNAs) (*Jinek et al., 2012*; *Mali et al., 2013*; *Hilton et al., 2015*; *Stepper et al., 2017*; *Kwon et al., 2017*; *Li et al., 2021*). CRISPR-Cas9-based epigenome editing strategies facilitate unprecedented, precise control of the epigenome and gene activation, providing a path to epigenetic-based therapeutics (*Cheng et al., 2019*).

A major challenge for epigenome editing is designing gRNAs that can achieve a desired level of transcriptional or epigenetic modulation. Finding effective gRNAs currently typically requires expensive and low-throughput experimental strategies (*Mohr et al., 2016*; *Liu et al., 2020*; *Mahata et al., 2023*). An alternative approach would be to computationally model how epigenome editing impacts histone PTMs as well as how perturbing these PTMs would consequently impact gene expression.

To understand how histone PTMs relate to gene expression, large epigenetic and transcriptomic datasets are required. Advancements in high-throughput sequencing have allowed quantification of gene expression and profiling of histone PTMs. Large consortia have performed an extensive number of assays across a wide variety of cell types (*The ENCODE Project Consortium, 2012*; *Kundaje et al., 2015*; *Barrett et al., 2012*).

These include measurements of histone PTMs, transcription factor binding, gene expression, and chromatin accessibility. These data have enhanced our understanding of how histone PTMs and other chromatin dynamics impact transcriptional regulation (*Keung et al., 2015*; *Rao et al., 2014*; *Holoch and Moazed, 2015*).

Studying the function of these histone PTMs, however, has been largely limited to statistical associations with gene expression, which may not capture causal relationships (*Karlić et al., 2010*; *Stillman, 2018*; *Singh et al., 2016*). For example, deep learning has been successful in predicting gene expression from epigenetic modifications, such as transcription factor binding (*Schmidt et al., 2017*), chromatin accessibility (*Schmidt et al., 2020*), histone PTMs (*Singh et al., 2016*; *Sekhon et al., 2018*; *Frasca et al., 2022*; *Singh et al., 2017*; *Hamdy et al., 2022*; *Chen et al., 2022*), and DNA methylation (*Zhong et al., 2019*). However, these studies predict gene expression as binary levels instead of a continuous quantity. Finally, as statistical associations can be driven by non-causal mechanisms, it is unclear whether such computational models learn mechanistic, causal relationships between various epigenetic modifications and gene expression. Beyond modeling the relationship between histone PTMs and gene expression, to fully describe how a particular gRNA would affect gene expression, a model of how epigenome editing affects histone PTMs is also required. To our knowledge, there currently are no computational models that can accurately model, in silico, the impact of epigenome editing on histone PTMs.

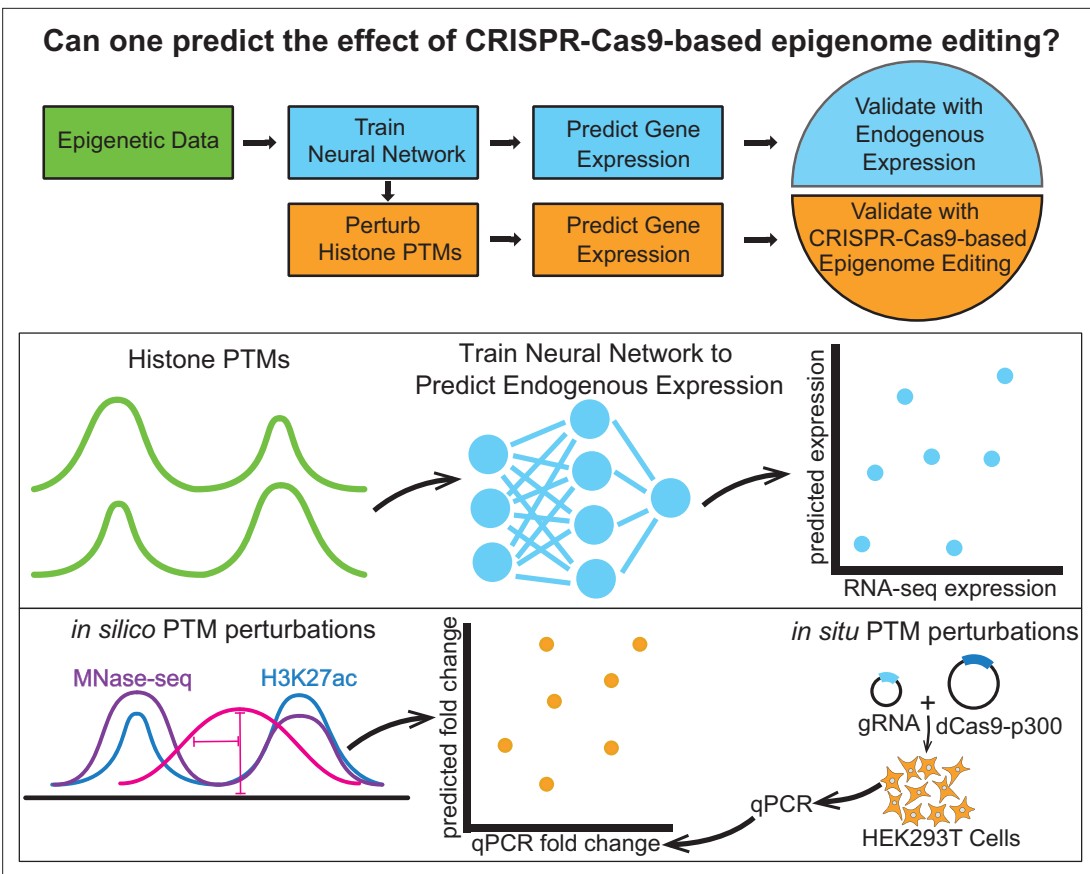

**Figure 1.** Schematic of the epigenome editing prediction pipeline. The pipeline uses epigenetic data to train models to predict endogenous gene expression. These models were used to predict fold-change in gene expression based on perturbed histone PTM input data, and their predictions were validated using CRISPR-Cas9-based epigenome editing data.

Motivated by these observations, we explored models for how epigenome editing impacts histone PTMs as well as how histone PTMs impact gene expression. We used data available through ENCODE (*Schreiber et al., 2020a*; *The ENCODE Project Consortium, 2012*) to train a model of how histone PTMs impact gene expression. Our model is highly predictive of endogenous expression and learns an understanding of chromatin biology which is consistent with known patterns of various histone PTMs (*Kimura, 2013*). To test this model in the context of epigenome editing, we generated perturbation data using the dCas9-p300 histone acetyltransferase system (*Hilton et al., 2015*). The dCas9-p300 system is thought to act primarily through local acetylation of histone lysine residues, particularly histone subunit H3 lysine residue 27 (H3K27ac). Therefore, we modeled the impact of dCas9-p300 on the epigenome as a local increase in the H3K27ac profile near the target site; since the precise effect of these perturbations is unknown, we tried a variety of potential modification patterns. We then applied our trained model to predict the impact of these putative H3K27ac modifications on gene expression (*Figure 1*). We found that our models, which are designed to predict gene expression values, were effective in ranking relative fold-changes among genes in response to the dCas9-p300 system, achieving a Spearman's rank correlation of ~0.8. However, their performance in ranking fold-changes within individual genes was less successful when compared to the prediction of gene expression across cell types from their native epigenetic signatures. We offer possible explanations in the discussion section.

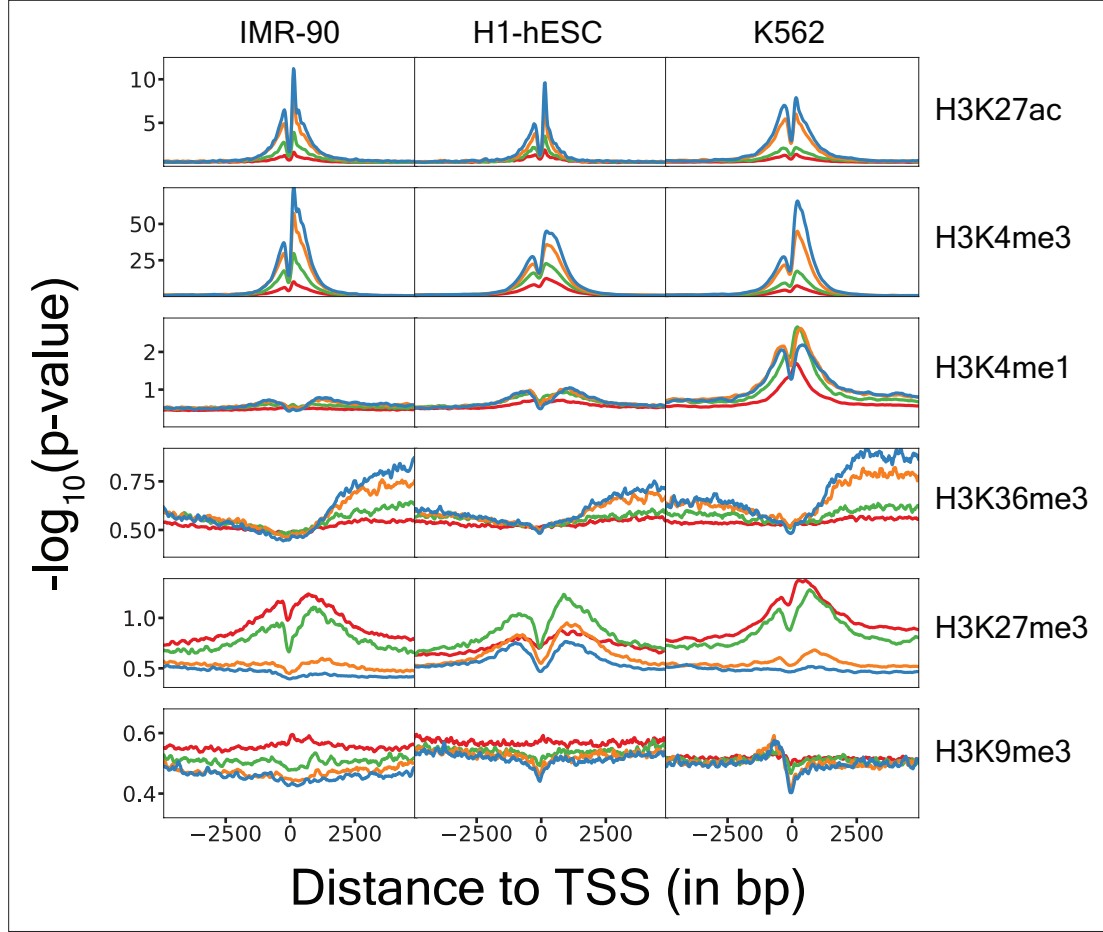

**Figure 2.** Metagene plots show histone post-translational modifications (PTMs) are consistent across cell types and recapitulate established relationships between histone PTMs and gene expression. Colors represent genes binned into quantiles based on gene expression. Blue 75–100%, orange 50–75%, green 25–50%, and red 0–25% of gene expression within a cell type. The $y$-axis represents $-\log_{10}(p\text{-value})$ obtained from ChIP-seq data.

The online version of this article includes the following figure supplement(s) for figure 2:

**Figure supplement 1.** Metagene plots for different cell types for uncorrected ChIP-seq data across gene expression quantiles.

**Figure supplement 2.** S3norm-based approach for correcting ChIP-seq $-\log_{10}(p\text{-values})$.

**Figure supplement 3.** Metagene plots for different cell types for batch effect-corrected ChIP-seq data across gene expression quantiles.

## Results

### Histone PTM data are highly predictive of gene expression

Genome-scale datasets are required to train models to predict gene expression using histone PTMs. Therefore, we obtained histone PTM ChIP-seq and RNA-seq data for 13 different human cell types from ENCODE (*Schreiber et al., 2020a*; *The ENCODE Project Consortium, 2012*; *Appendix 1— table 1*). We inspected metagene plots (histone PTMs averaged across genes within gene expression quantiles) describing 6 histone PTMs in each of these 13 different cell types. Based on different overall signal levels across cell types, we concluded that batch effects, likely due to inconsistent sequencing depths, would need to be corrected prior to training models (*Figure 2—figure supplement 1*).

We corrected these batch effects by adapting S3norm (*Xiang et al., 2020*; 'Materials and methods', *Figure 2—figure supplement 2*). These corrected histone PTM tracks were then used for the remainder of our analyses along with RNA-seq data for each of the 13 cell types (*Figure 2—figure supplement 3*).

Importantly, we observed that H3K27ac and H3K4me3 histone PTM signal strengths positively covaried with gene expression quantile (representative cell types shown in *Figure 2*; all cell types

shown in *Figure 2—figure supplement 3*). Conversely, repressive histone PTMs such as H3K27me3 and H3K9me3 were strongly inversely correlated with gene expression quantiles. Spatial patterns in the metagene plots for H3K36me3 suggested that this mark covaried more strongly with gene expression in the gene body than near the TSS. Taken together, these observations recapitulated the current understanding of these well-studied histone PTMs with respect to their associations to gene expression (*Kimura, 2013*; *Millán-Zambrano et al., 2022*; *Zhao et al., 2021*).

## Histone PTMs accurately predict endogenous gene expression

To predict how epigenome editing affects gene expression, we first trained models to predict gene expression from endogenous histone PTMs. We trained several convolutional neural networks (CNNs) and ridge regression models to predict the gene expression of each gene in each of the 13 cell types, using only histone PTM data proximal to the TSS as features ('Materials and methods', *Figure 2*). We observed that Spearman's rank correlation between the true gene expression and the models' predicted gene expression on held-out chromosomes improves as the input context size increases; and for all input context sizes, the CNNs outperform ridge regression models (*Figure 3A*). Therefore, for the remainder of the analyses, we use a context size of 10,000 base pairs.

To assess the models' ability to generalize to unseen cell types, we trained a set of 10 models for each cell type. In particular, we held out the histone PTMs for a given cell type during training and then tested the models on that held-out cell type.

We observed that the CNNs outperformed ridge regression models on this cross-cell type generalization task across essentially all cell types (*Figure 3B*). The reduced performance on the adrenal cell type may be driven by a cell-type-specific biological mechanism that leads to a lower correlation of its epigenetic data with other cell types, particularly for H3K36me3 (*Figure 3—figure supplement 1*).

Although our models accurately predicted endogenous gene expression, this does not guarantee their ability to accurately predict the relationship between local histone PTM variations and gene expression for a particular gene across different cell types. Therefore, we determined Spearman's rank correlations between the observed expression and the predicted expression for each held-out gene across the different cell types. The distribution of these correlations suggests that overall the CNNs can better rank cell types by gene expression than ridge regression (*Figure 3C*). In particular, the median cross-cell type correlation is ~0.53 for CNNs compared to ~0.39 for ridge regression.

We also benchmarked the predictive performance of our CNN model against existing methods and observed that we outperform all existing methods across multiple cell types (*Figure 3—figure supplement 3*).

## Models recover established relationships between histone PTMs and gene expression

We investigated what features of the data the models used to predict gene expression. For a given gene, we modified the input histone PTMs one-by-one at nucleosome-scale and measured the predicted fold-change in gene expression (*Figure 4*, *Figure 4—figure supplement 1*, 'Materials and methods').

We observed considerable changes to the predicted fold-change upon modifying different histone PTMs. In particular, our CNN models predict that repressive marks such as H3K27me3 and H3K9me3 proximal to the TSS result in a slight decrease in expression. In contrast, activating histone PTMs such as H3K27ac and H3K4me3 results in an almost twofold increase in predicted gene expression near the TSS. Activating both of these markers exhibits a periodic pattern, likely reflecting nucleosome occupancy. However, activation of H3K4me3 results in a sharp increase in gene expression downstream of the TSS. Additionally, we observed that H3K36me3 is predicted to increase expression, but only if it is deposited in the gene body, and the degree of activation gradually increases as it is deposited further inside of the gene body. The consistency of these observations with established mechanisms, observed previously in the literature, via which these histone PTMs modulate gene expression (*Kimura, 2013*) lends credence to our gene expression models and shows that these models learn the spatial patterns of histone PTMs.

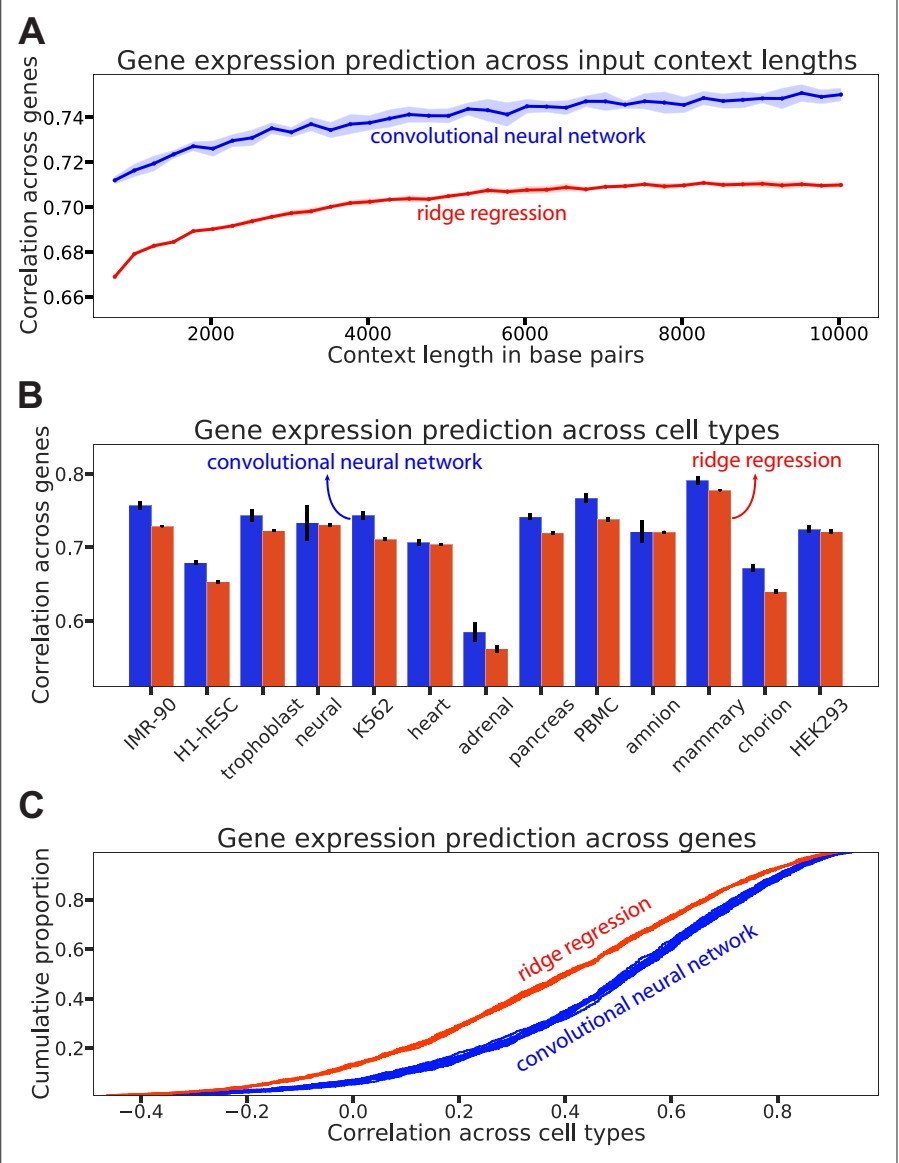

**Figure 3.** Histone post-translational modifications (PTMs) accurately predict endogenous gene expression. (**A**) Spearman correlation on genes from held-out chromosomes for different input context lengths, with all cell types pooled together. The blue curve is the mean across 10 computational replicates of convolutional neural networks (CNNs), and the red is the mean across 10 computational replicates of ridge regression. Shaded area represents standard deviation in the Spearman correlation across the 10 computational replicates. (**B**) Spearman correlation on genes of cell types held out during training. The bar plots represent the mean across 10 computational replicates, and the error bars represent the corresponding standard deviations. (**C**) Distribution of Spearman correlations across genes, computed for each gene in test chromosomes by comparing predictions across the 13 cell types. The different curves represent 10 computational replicates for each model type.

The online version of this article includes the following figure supplement(s) for figure 3:

**Figure supplement 1.** Spearman correlation distribution across all cell types, for each cell type.

**Figure supplement 2.** Endogenous RNA-seq expression levels of HEK293 and HEK293T cell lines are highly concordant.

**Figure supplement 3.** Benchmarking of models predicting gene expression from histone marks.

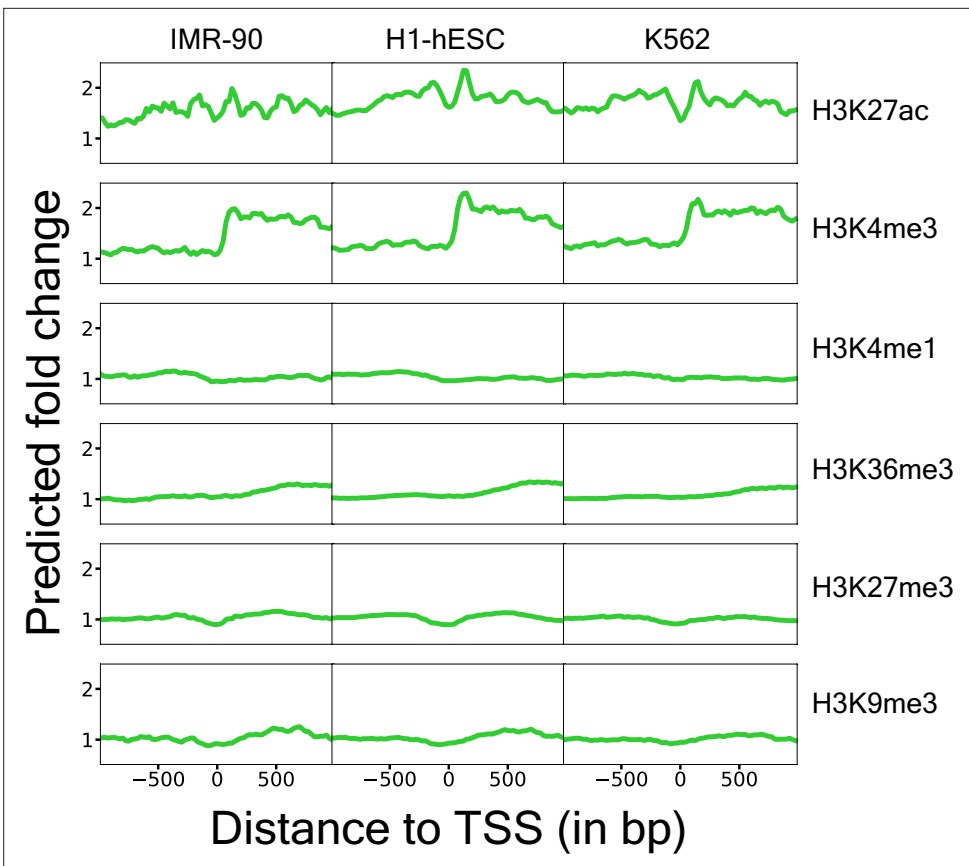

**Figure 4.** Features learned by gene expression models. Each point on the X-axis corresponds to in silico perturbation of that assay at that position, and the Y-axis measures the predicted fold-change in gene expression, averaged across a set of 100 trained models. The fold-changes were averaged across 500 randomly chosen genes.

The online version of this article includes the following figure supplement(s) for figure 4:

**Figure supplement 1.** Features learned by gene expression models for H3K9me3 in K562.

## dCas9-p300 differentially activates genes depending on gRNA-targeted site

To test if our gene expression models could accurately predict the outcome of in situ epigenome editing experiments, we first generated dCas9-p300 data in the HEK293T cell line for eight genes (*Figure 5*). We assayed at least five gRNAs per gene with at least three replicates for each gRNA. We used the HEK293T cell line because it is a widely used testbed for epigenome editing strategies (*Hilton et al., 2015*; *Nuñez et al., 2021*; *O'Geen et al., 2017*; *Mahata et al., 2023*; *Escobar et al., 2022*; *Wang et al., 2022*). Based on *Figures 2 and 4*, the largest changes in H3K27ac across gene expression quantiles occur within 500 base pairs of the TSS, so we constrained gRNA targeting to this critical window. We filtered gRNAs for predicted specificity (*Concordet and Haeussler, 2018*) and on-target activity scores (*Sanson et al., 2018*). Each gRNA was tested individually, and relative mRNA abundance was measured using quantitative PCR (qPCR).

We successfully increased gene expression of all eight genes with fold-change activation using the most effective respective gRNA for each gene ranging from 3-fold to ~6500-fold relative to a non-targeting control gRNA (*Figure 5C*). Some of this variation may be explained by differences in endogenous gene expression levels, with the targeting of lowly expressed genes resulting in higher fold-change measurements (*Appendix 1—table 2*), as observed previously (*Wang et al., 2022*). Nevertheless, substantial variability was observed in gRNA efficacy for all targeted genes. In particular, two (*MYO1G* and *PRSS12*) out of eight genes had the most efficacious gRNA downstream of the TSS. This contrasts with other reports where targeting CRISPR/Cas-based activators upstream of the TSS leads to the highest activation (*Mohr et al., 2016*; *Gilbert et al., 2014*).

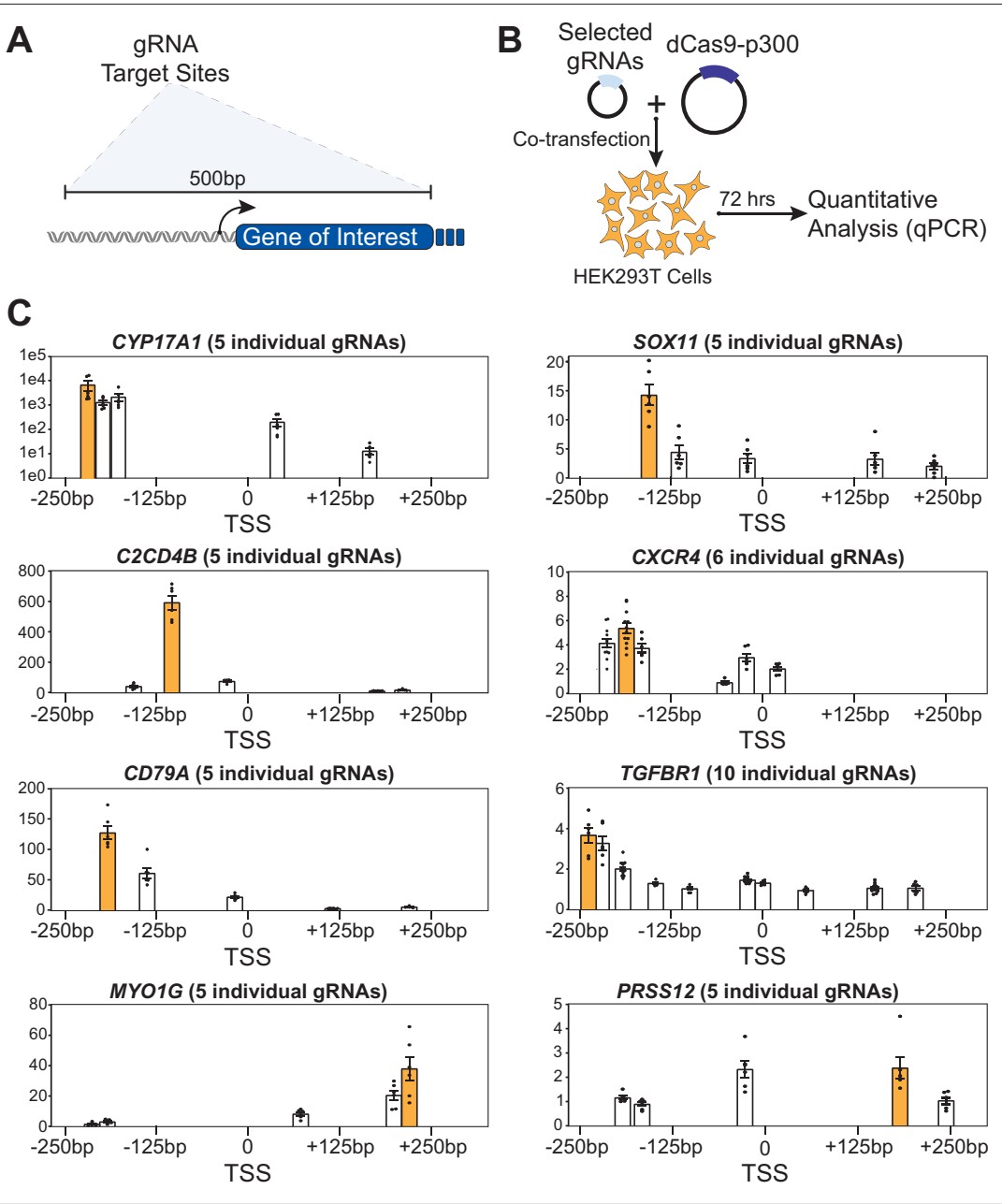

**Figure 5.** dCas9-p300 epigenome editing at eight endogenous genes identifies gene-specific responses. The genes tested are *CYP17A1*, *SOX11*, *C2CD4B*, *CXCR4*, *CD79A*, *TGFBR1*, *MYO1G*, and *PRSS12*. (**A**) gRNA (n=5) targeting +/− 250 bp of each gene was selected. (**B**) These selected gRNA were individually co-transfected with dCas9-p300 with relative mRNA determined with qPCR. (**C**) Relative mRNA associated with selected guide position is displayed with the highest activating guide position marked in orange. The Y-axis corresponds to qPCR fold-change.

The online version of this article includes the following figure supplement(s) for figure 5:

**Figure supplement 1.** Transfection efficiency is shared across experiments.

These data indicate that the rules governing the outcomes for successful dCas9-p300-based epigenome editing – and subsequent increased transcriptional activation – are complex and highlight the fact that locus-specific nuances can be important factors in epigenome editing experiments. For example, two gRNAs targeting within ~50 base pairs of each other on *C2CD4B* have a 100-fold difference in measured mRNA (*Figure 5C*). Further, gRNAs targeting the same position in different genes

can have vastly different effects. For instance, several gRNAs targeting ~250 base pairs upstream of the *CYP17A1* TSS result in a high fold-change, while two gRNAs targeting roughly the same position in *MYO1G* failed to produce substantial activation (*Figure 5C*).

## Computationally predicting the outcome of dCas9-p300 epigenome editing experiments

To test the hypothesis that dCas9-p300 acts through the local deposition of H3K27ac, we modeled this process in silico and used these perturbations as inputs to our models trained on endogenous gene expression.

We modeled the effect of dCas9-p300 on histone PTMs based on evidence from the literature as well as additional experiments we performed. The key assumptions of this model are (1) there exists steric hindrance of dCas9 by nucleosomes (*Makasheva et al., 2021*; *Horlbeck et al., 2016*; *Isaac et al., 2016*; *Radzisheuskaya et al., 2016*); (2) dCas9-p300 acts locally, altering H3K27ac levels near the gRNA target locus (*Gemberling et al., 2021*; *Dominguez et al., 2022*) (we adopted this simplifying assumption since off-target effects are unpredictable and underexplored; *Dominguez et al., 2022*; *Gemberling et al., 2021*; *Weinert et al., 2018*); and (3) dCas9-p300 can deposit H3K27ac at nucleosomes, as defined by MNase activity (see 'Materials and methods'; *Segelle et al., 2022*; *Zhou et al., 2016*). Our resulting in silico perturbation model had a number of free parameters that we briefly describe below. Wherever possible, we used values for these parameters obtained from the literature or tested a range of plausible values. For a more complete description of the model, see 'Materials and methods'.

The first component of our perturbation model is steric hindrance of dCas9-p300 by nucleosomes (*Figure 6A*). Intuitively, if DNA is tightly wound around a nucleosome, the gRNA would be less likely to bind successfully. Mathematically, we modeled this as an inverse relationship between the amount of H3K27ac deposited and the MNase activity at the gRNA target locus.

It is widely assumed that dCas9-p300 activates genes through the local deposition of H3K27ac (*Klann et al., 2017*; *Dominguez et al., 2022*). To model this, we increased local levels of H3K27ac relative to endogenous levels according to a Gaussian kernel centered at the gRNA target locus (*Figure 6B*). This adds acetylation primarily within a distance controlled by the standard deviation ($\sigma$) of the kernel. We performed CUT&RUN experiments (see Appendix 1) that suggest that this distance is at least 1000 base pairs (*Figure 6—figure supplement 1*). Since we also do not know the degree to which dCas9-p300 alters H3K27ac levels, we modeled this as another free parameter, $\lambda$, which we varied over a range of plausible values ('Materials and methods').

Finally, we assumed that dCas9-p300 does not affect the positioning of nucleosomes and hence can only add H3K27ac at positions currently occupied by histones (*Zhou et al., 2016*). As such, we expect H3K27ac levels to only increase at loci where there is MNase activity. In particular, we modulated the Gaussian kernel described above by performing point-wise multiplication with MNase activity (*Figure 6C*).

Since nucleosome positioning plays a crucial role in our perturbation model, we generated, to our knowledge, the first MNase-seq data for the HEK293T cell line (see Appendix 1).

To get a baseline of how well our perturbation model might be able to predict the effect of dCas9-p300 on gene expression, we considered the 13 distinct cell types as being analogous to natural perturbations of local histone PTMs. Across the eight genes discussed above, which were excluded from the training set, we observed a Spearman's rank correlation of ~0.8 between the endogenous expression and that predicted by our expression model (*Figure 6D*). This correlation was in line with the correlation observed across the endogenous transcriptome (*Figure 3A and B*). We further observed that our expression models were able to accurately rank gene expression across cell types within individual genes (*Figure 6—figure supplement 2*).

We then computed fold-changes between the expression predicted using endogenous histone PTMs and the expression predicted using in silico perturbations of these histone PTMs. We observed that our models were effective in ranking relative fold-changes across genes in response to dCas9-p300, achieving a Spearman's rank correlation of ~0.8 between these predicted fold-changes and the experimentally determined mRNA fold-changes induced by dCas9-p300 (*Figure 6E*). However, the performance in ranking fold-changes within individual genes was less accurate (*Figure 6—figure supplement 3*) when compared to the prediction of cell-type-specific gene expression from native epigenetic signatures (*Figure 6—figure supplement 2*).

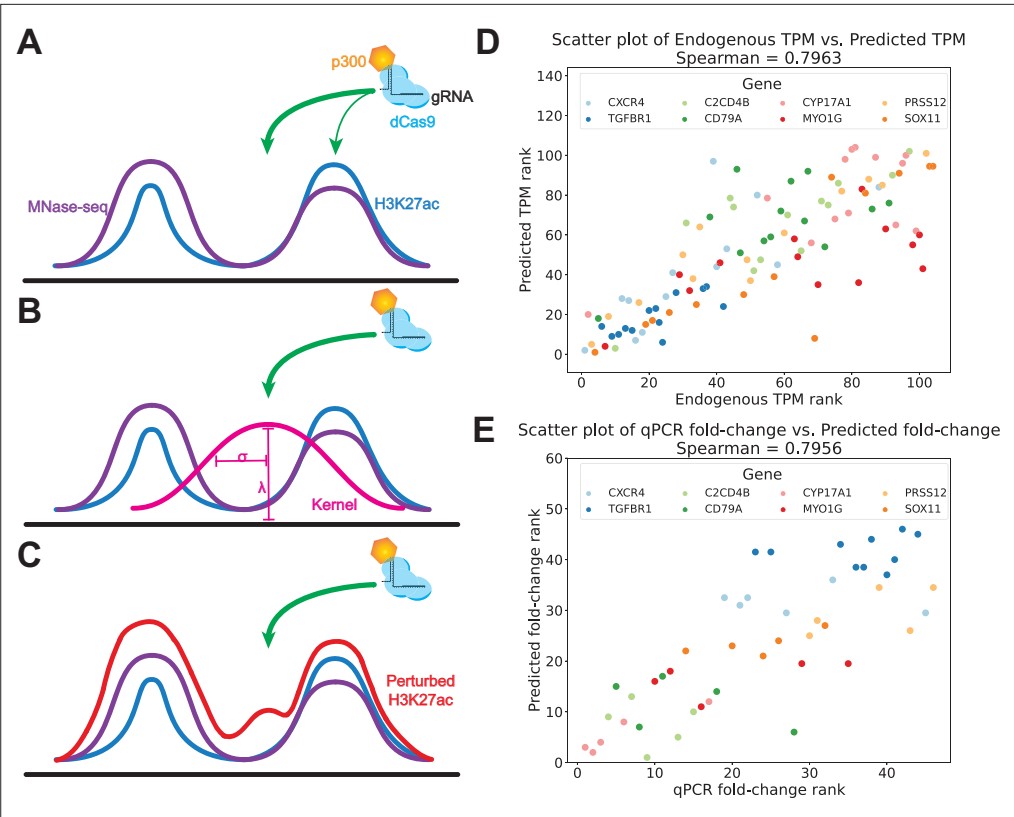

**Figure 6.** In silico model for dCas9-p300-based epigenome editing. (**A**) dCas9-p300 is more likely to bind to a position not occupied by the nucleosome. Thicker green arrow represents higher probability of binding for a gRNA targeting that site. (**B**) The in silico perturbation is modeled as a Gaussian kernel parameterized by a standard deviation, $\sigma$, and the amount of H3K27ac deposited, $\lambda$. (**C**) The final perturbed H3K27ac is obtained by point-wise multiplication of the Gaussian kernel with nucleosome occupancy quantified by MNase activity since dCas9-p300 can only acetylate histones within nucleosomes. (**D**) Ranks for predicted and endogenous expression across 8 genes and 13 cell types. Rank 1 corresponds to the highest numerical value. (**E**) Ranks for predicted and empirically measured expression fold-changes following perturbation by dCas9-p300 for eight genes in HEK293T cells. Rank 1 corresponds to the highest numerical value.

The online version of this article includes the following source data and figure supplement(s) for figure 6:

**Source data 1.** Raw qPCR data.

**Source data 2.** Raw CUT&RUN qPCR data.

**Source data 3.** Primer sequences, sources, assay use, and corresponding direction.

**Figure supplement 1.** H3K27ac levels elevation is similar across quantified regions following gRNA dCas9-p300 targeting.

**Figure supplement 2.** Gene-wise predicted vs. experimental gene expression transcripts per million (TPM) ranks.

**Figure supplement 3.** Gene-wise predicted vs experimental fold-change ranks.

**Figure supplement 4.** Predicted vs. experimental fold-change ranks.

We extended this analysis to a Perturb-seq dataset, consisting of gRNAs targeting proximal to the TSS of five genes in the K562 cell line to further assess the model's ability to estimate gene expression changes. Consistent with the performance observed in *Figure 6E*, the model demonstrated robustness in predicting gene expression fold-changes across these 27 gRNAs targeting these five genes. Notably, these predictions achieved a Spearman's rank correlation of ~ 0.47 with the experimentally determined mRNA fold-changes measured by Perturb-seq, as shown in *Figure 6—figure supplement 4* (see Appendix 1). These results reinforce the model's effectiveness in capturing the nuanced effects of epigenome editing across different genes and cell types.

## Discussion

Here, we sought to investigate whether we could predict how targeted epigenome editing affects endogenous gene expression. First, we collected data from ENCODE which reflects how PTMs to histones covary with gene expression across cell types. We trained models to predict endogenous gene expression from these histone PTMs and found that these models were highly predictive (*Figure 3*). We further showed that such models learned known relationships between histone PTMs and gene expression (*Figure 4*). To test whether these expression models could predict the outcomes of epigenome editing experiments, we generated dCas9-p300 epigenome editing data in the HEK293T cell line for eight genes along with genome-wide MNase-seq data for this testbed cell line. We anticipate that the genome-wide nucleosome occupancy information for the HEK293T cell line provided by our MNase-seq experiment will be a useful resource for the genomics community. We also generated dCas9-p300 epigenome editing data via a Perturb-seq experiment in K562 cells with gRNAs targeting the promoter regions of five genes. In this study, we focused on the histone changes induced by dCas9-p300 epigenome editing, but future studies may use the framework described in our manuscript and apply it to other transcriptional editors as well.

We modeled dCas9-p300's impact on local H3K27ac using a variety of parameter choices and found that these models accurately predicted fold-changes across genes. However, they were less accurate at predicting the outcome of these experiments within a given gene, as compared to predicting gene expression from the endogenous epigenetic signatures (*Figure 6—figure supplements 2 and 3*). Since the endogenous epigenetic signatures could be different across genes, these *global* factors might drive the models' accurate inter-gene fold-change prediction accuracy. However, since ranking fold-changes within a gene requires a detailed understanding of the epigenetic profiles before and after dCas9-p300 epigenome editing, the reduction in performance from predicting endogenous expression to predicting the outcome of epigenome editing experiments is likely explained by one or more of the following hypotheses: (1) dCas9-p300 activates gene expression by mechanisms other than the *local* acetylation of H3K27 or dCas9-p300 functions differently from native p300; (2) differences in gRNA efficacy are not accurately explained by existing computational scores; or (3) our models, trained on endogenous gene expression across various cell types, failed to generalize even if dCas9-p300 perturbations are correctly modeled. We discuss these possible explanations more in depth below.

We considered numerous models of how dCas9-p300 affects local histone PTMs. These models span current hypotheses of how dCas9-p300 alters local histone PTMs such as H3K27ac. The poor generalization of our models in predicting intra-gene epigenome editing fold-changes could be explained by dCas9-p300 acting via mechanisms beyond local acetylation of histone proteins and H3K27 (*Zhao et al., 2021*). For example, p300 is a promiscuous lysine acetyltransferase and dCas9-p300 could be broadly acetylating across the proteome impacting *trans* factors (*Weinert et al., 2018*). Alternatively, local acetylation could be contingent on unmodeled factors such as *trans*-acting proteins or other histone PTMs present at the locus (*Zhao et al., 2021*; *Zheng et al., 2021*). Furthermore, the genome-wide specificity of dCas9-p300-mediated histone acetylation – although likely better than small molecule-based perturbations – remains imperfect (*Gemberling et al., 2021*; *Dominguez et al., 2022*). Our inability to accurately predict the relative fold-change of different gRNAs targeting the same gene suggests that these unmodeled factors would have to differentially affect neighboring loci within the same gene. This highlights that the current understanding of the mechanism via which dCas9-p300 drives gene expression is potentially incomplete. To better understand this mechanism, it would be immensely helpful to generate a compendium of histone PTM profiles before and after performing epigenome editing, which would enable us to train better machine learning models to predict the impact of dCas9-p300 on gene expression.

Another possible explanation for the drop in accuracy is varying gRNA efficacies. For example, gRNAs might have different levels of on-target and off-target effects. Although we ensured that all of the gRNAs used in generating the dCas9-p300 epigenome editing data were predicted to have high on-target and low off-target scores, we observed examples of gRNAs that targeted roughly the same genomic position but had vastly different impacts on gene expression. This suggests that these differences could be driven by inconsistencies in gRNA efficacy instead of local acetylation dynamics. Generating a large number of pairs of gRNAs, such as through CRISPR screens (*Schmidt et al., 2022*), targeting nearby positions could help to elucidate the factors that drive differential gRNA efficacy for epigenome editing.

The ambiguity in how to accurately model the impact of epigenome editing stands in contrast to the simpler case of DNA sequence changes, where perturbations are relatively trivial to model. Indeed, dCas9-p300 changes histone PTMs in complex ways, rendering the modeling of such perturbations much more challenging. In contrast, models like Enformer (*Avsec et al., 2021*) that predict gene expression directly from DNA sequence may be able to generalize to DNA sequence perturbations better due to their relative simplicity.

Another source of generalization error could be extrapolating beyond the range of the training data. Massively increasing the amount of H3K27ac at a locus may make a gene look different than any other endogenous gene observed during training. Regression approaches including neural networks are known to have limitations in extrapolation (*Xu et al., 2020*).

Our research indicates that we can predict endogenous gene expression accurately based on histone PTMs. By creating a comprehensive dataset of epigenome editing, which assays histone PTMs before and after in situ perturbations, we can enhance machine learning models. This will improve our understanding of the effects of dCas9-p300 on gene expression and assist in the design of gRNAs for achieving fine-tuned control over gene expression levels. These advancements are vital for devising experiments that deepen our mechanistic insight and offer effective strategies for human epigenome editing.

## Materials and methods
### Data preparation
We obtained $-\log_{10}(p\text{-value})$ ChIP-seq tracks created by running the MACS2 peak-caller (*Feng et al., 2012*) on read count data, from the ENCODE Imputation Challenge (*Schreiber et al., 2020a*). For three tracks where data were not available, we downloaded Avocado (*Schreiber et al., 2020b*) imputations from the ENCODE data portal (*The ENCODE Project Consortium, 2012*). We binned each epigenetic track at 25 base pair resolution and pre-processed them with an additional log operation before inputting them into the models for training.

We downloaded polyA-plus RNA-seq gene expression transcripts per million (TPM) values for each of the 13 cell types in *Appendix 1—table 1*, from the ENCODE data portal (*The ENCODE Project Consortium, 2012*) and preprocessed them with a log operation.

### Normalizing *p*-values by adapting S3norm
We assigned IMR-90 to be a reference cell type, for each of the six histone PTMs and kept its *p*-values unchanged. We then performed a transformation for each of the remaining cell types adapted from the core technique developed by S3norm (*Xiang et al., 2020*), in order to normalize each histone PTM track in each of these remaining cell types, with respect to the corresponding histone PTM track in IMR-90.

First, we computed *peaks* in both the reference as well as the target cell type. *Peaks* were defined as the 25 base pair bins corresponding to FDR-adjusted *p*-values <0.05 (*Benjamini and Hochberg, 1995*). For histone PTM tracks that were obtained from Avocado imputations (due to lack of availability of experimental data), *peaks* were defined to be the 1000 bins containing the smallest Avocado imputed *p*-values (based on suggestions from the authors of Avocado *Schreiber et al., 2020b*). All the remaining bins were defined to be *background*, for both the reference as well as the target cell types.

We then computed the list of *peaks* that were common to both the reference and the target cell types. These were termed *common peaks*. Similarly, we defined *common background* as the list of bins that were assigned to be *background* in both the reference as well as the target cell types.

The S3norm method was designed to work with count data, which is always $\geq 1$. However, the histone PTM tracks, which are represented as $-\log_{10}(p\text{-values})$, are not guaranteed to always be $\geq 1$; hence, we transformed all the histone PTM tracks by adding 1 to the $-\log_{10}(p\text{-values})$, in both the reference as well as the target cell types.

Additionally, since the histone PTM tracks obtained from imputations performed by Avocado were not guaranteed to be distributed similar to experimental $-\log_{10}(p\text{-values})$, we scaled all the histone PTM tracks (both experimental as well as Avocado imputations) by dividing them by the minimum observed value in *common peaks* and *common background*, in order to bring experimental data and

Avocado imputations onto a similar footing. In particular, before applying the S3norm normalization, we transformed $-\log_{10}(p\text{-values})$ in *common peaks* and *common background* for both the reference as well as the target cell type as follows:

$$\text{TransformedCommonPeaks}_{i,\text{reference}} = \frac{1 + \text{CommonPeaks}_{i,\text{reference}}}{\min_i(\text{CommonPeaks}_{i,\text{reference}})} \tag{1}$$

$$\text{TransformedCommonPeaks}_{i,\text{target}} = \frac{1 + \text{CommonPeaks}_{i,\text{target}}}{\min_i(\text{CommonPeaks}_{i,\text{target}})} \tag{2}$$

$$\text{TransformedCommonBackground}_{i,\text{reference}} = \max(\frac{1 + \text{CommonBackground}_{i,\text{reference}}}{\min_i(\text{CommonBackground}_{i,\text{reference}})}, 0) \tag{3}$$

$$\text{TransformedCommonBackground}_{i,\text{target}} = \max(\frac{1 + \text{CommonBackground}_{i,\text{target}}}{\min_i(\text{CommonBackground}_{i,\text{target}})}, 0) \tag{4}$$

The normalization procedure of S3norm then wishes to find two positive parameters, $\alpha$ and $\beta$ that are to be learned from the data such that both the following equations are satisfied:

$$\text{mean}(\text{TransformedCommonPeaks}_{\text{reference}}) = \text{mean}(\alpha \times \text{TransformedCommonPeaks}^{\beta}_{\text{target}}) \tag{5}$$

$$\text{mean}(\text{TransformedCommonBackground}_{\text{reference}}) = \text{mean}(\alpha \times \text{TransformedCommonBackground}^{\beta}_{\text{target}}) \tag{6}$$

Specifically, $\alpha$ is a scale factor that shifts the transformed $-\log_{10}(p\text{-values})$ of the target data set in log scale, and $\beta$ is a power transformation parameter that rotates the transformed $-\log_{10}(p\text{-values})$ of the target data set in log scale (*Figure 2—figure supplement 2*). There is one and only one set of values for $\alpha$ and $\beta$ that can simultaneously satisfy both the above equations for *common peaks* and the *common background* (*Xiang et al., 2020*).

The values of $\alpha$ and $\beta$ were estimated by the Powell minimization method implemented in scipy (*Fletcher and Powell, 1963*; *Virtanen et al., 2020*). The resulting normalized $-\log_{10}(p\text{-values})$ were used for all downstream analyses (*Figure 2—figure supplement 3*).

## Training endogenous gene expression models

We trained CNN and ridge regression models, each, to predict gene expression using histone PTM tracks. Input features for each gene were centered at its TSS. We used an input context size of 10,000 base pairs for all analyses subsequent to *Figure 3*. For all analyses, we obtained predictions from our models by averaging predictions ensembled across 100 computational replicates.

To train CNN models, the normalized histone PTM tracks for each gene were processed with successive convolutional blocks. Each convolutional block consisted of a batch-normalization layer, rectified linear units (ReLU), a convolutional layer consisting of 32 convolutional kernels, each of width 5, followed by a dropout with 0.1 probability. Finally, a pooling layer was applied to gradually reduce the dimension of the features. After being processed with five such convolutional blocks, the output was flattened and passed through a fully connected layer consisting of 16 neurons and a ReLU activation. This was ultimately processed with a fully connected layer with a single output and a linear activation (since this was a regression task). The models were trained with a mean squared error loss using the Adam optimizer with a learning rate of 0.001 for the first 50 epochs and 0.0005 for the remaining 50 epochs. Training CNN models took about 1.5 hours on 1 NVIDIA A100 Tensor Core GPU.

## Interrogating the features learned by CNNs

To see how different features affected predicted levels of expression, we systematically perturbed each input feature and determined how much the perturbation affected predicted expression levels. To be concrete, we denoted the epigenetic feature at position $i$ of gene $g$ in cell type $CT$ as $E_i^{CT,g}$. We then defined a perturbation function that added a scalar value of $\lambda_0 = 2500$ to the epigenetic features within 3 bins of a focal position, say, $j$:

$$F_j\left(E_1^{CT,g}, \ldots, E_W^{CT,g}\right) := \left(E_1^{CT,g}, \ldots, E_{j-3}^{CT,g} + \lambda_0, \ldots, E_{j+3}^{CT,g} + \lambda_0, \ldots, E_W^{CT,g}\right),$$

recalling that $W$ is the number of bins of 25 base pairs considered by our models, which is set to 401, corresponding to a 10,000 base pair input context length, for all analyses subsequent to *Figure 3*.

These perturbations corresponded to ~150 base pairs, which is roughly the length of DNA wrapped around a nucleosome.

To produce *Figure 4*, we applied the above perturbation functions, $F_1, \ldots, F_W$ to a histone PTM track of interest, and then measured the fold-change in predicted expression. To account for differences in the endogenous histone PTM tracks between genes, we averaged these fold-changes across 500 randomly chosen genes.

### In silico modeling of dCas9-p300-based epigenome editing

Our model of how dCas9-p300 perturbs local histone PTMs has three separate components. We describe each of these components in turn, and then present the full model below. Throughout, we write $j$ for the position that the gRNA targets.

First, we modeled steric hindrance of dCas9 due to nucleosomes. We used MNase-seq signal strength as a proxy for nucleosome occupancy. Letting $m_j$ be the MNase-seq read coverage at the gRNA binding site, we modeled steric hindrance by scaling the acetylation activity of dCas9-p300 by a factor of $\exp\left(-5 \times m_j\right)$.

Second, we assumed that dCas9-p300 primarily alters the levels of H3K27ac only locally. As such, we modeled the acetylation activity of dCas9-p300 at a particular locus as a Gaussian kernel centered at the gRNA. Concretely, the acetylation activity at position $i$ is multiplied by a factor of $\exp\left(-(i-j)^2/2\sigma^2\right)$, where $\sigma^2$ is a parameter of the model.

Finally, we assumed that dCas9-p300 can only acetylate histones where they currently are – it cannot move histones or increase H3K27ac levels outside of histones. To model this mathematically, we multiplied the acetylation activity at site $i$ by the MNase read coverage, $m_i$. Therefore, if the MNase read coverage is 0 (i.e., there is no evidence of histones at that locus), then the amount of H3K27ac added to that position is also 0.

Putting this all together, for a guide targeting at position $j$, the effect on H3K27ac levels at position $i$ is proportional to

$$\exp\left(-5\,m_j\right) \times \exp\left[\frac{-(i-j)^2}{2\sigma^2}\right] \times m_i$$

The constant of proportionality (i.e., how strong we expect dCas9-p300 to be overall) is treated as another free parameter, which we denote by $\lambda$.

ENCODE has epigenetic data for the HEK293 cell line, but we performed our dCas9-p300 perturbations in the HEK293T cell line. As such, we used the HEK293 histone PTM as well as RNA-seq data as a stand-in for the HEK293T histone PTM and RNA-seq levels. This substitution is justified as gene expression levels for HEK293 and HEK293T are highly concordant (*Figure 3—figure supplement 2*). Indeed, the Spearman's rank correlation between expression levels for HEK293 and two independent measurements of expression levels in HEK293T are 0.86 and 0.88, which are comparable to the correlation between the two independent experiments in HEK293T ($\rho = 0.92$). That is, the correlation across experiments within HEK293T cells is only slightly higher than the correlation between HEK293 and HEK293T, suggesting that cross-cell type differences between HEK293 and HEK293T are on the same order as the inherent experimental and biological noise within a single cell type.

### Experimental procedure

The details of dCas9-p300 epigenome editing, qPCR, CUT&RUN, and MNase-seq experiments are provided in the supplementary material.

### Acknowledgements

This research was supported in part by NIH grants R35-GM134922 and R35-GM143532.

# Additional information

## Funding

| Funder | Grant reference number | Author |
|---|---|---|
| National Institute of General Medical Sciences | R35-GM134922 | Yun S Song |
| National Institute of General Medical Sciences | R35-GM143532 | Isaac B Hilton |

The funders had no role in study design, data collection and interpretation, or the decision to submit the work for publication.

## Author contributions

Sanjit Singh Batra, Conceptualization, Data curation, Software, Formal analysis, Validation, Investigation, Visualization, Methodology, Writing – original draft; Alan Cabrera, Conceptualization, Resources, Data curation, Validation, Investigation, Visualization, Writing – original draft; Jeffrey P Spence, Conceptualization, Formal analysis, Investigation, Methodology, Writing – original draft; Jacob Goell, Selvalakshmi S Anand, Resources, Data curation, Investigation, Writing – review and editing; Isaac B Hilton, Conceptualization, Resources, Supervision, Funding acquisition, Investigation, Writing – original draft, Project administration, Writing – review and editing; Yun S Song, Conceptualization, Supervision, Funding acquisition, Investigation, Methodology, Writing – original draft

## Author ORCIDs

Jeffrey P Spence ⓘ https://orcid.org/0000-0002-3199-1447
Selvalakshmi S Anand ⓘ https://orcid.org/0009-0000-4436-2618
Isaac B Hilton ⓘ https://orcid.org/0000-0002-3064-8532
Yun S Song ⓘ https://orcid.org/0000-0002-0734-9868

Reviewer #1 (Public review): https://doi.org/10.7554/eLife.92991.4.sa1
Reviewer #2 (Public review): https://doi.org/10.7554/eLife.92991.4.sa2
Author response https://doi.org/10.7554/eLife.92991.4.sa3

# Additional files

## Supplementary files

MDAR checklist

## Data availability

The MNase-seq data for the HEK293T cell line is available at BioProject ID PRJNA892960 on SRA. The data from the dCas9-p300 K562 Perturb-seq experiments is available at GSE255610 on SRA. Code and data for training the gene expression models, along with code for generating the figures in the manuscript, are available at https://github.com/songlab-cal/epigenome_editing_2023 (copy archived at *Batra, 2023*).

The following dataset was generated:

| Author(s) | Year | Dataset title | Dataset URL | Database and Identifier |
|---|---|---|---|---|
| Batra SS, Cabrera A, Spence JP, Goell J, Anand SS, Hilton IB, Song YS | 2025 | MNase-seq data for the HEK293T cell line | https://www.ebi.ac.uk/ena/browser/view/PRJNA892960 | European Nucleotide Archive, PRJNA892960 |

The following previously published dataset was used:

| Author(s) | Year | Dataset title | Dataset URL | Database and Identifier |
|---|---|---|---|---|
| Goell J, Li J, Mahata B, Kim S, Shah S, Shah S, Contreras M, Misra S, Reed D, Bedford GC, Escobar M, Hilton IB | 2025 | Tailoring a CRISPR/Cas-based Epigenome Editor for Programmable Chromatin Acylation and Decreased Cytotoxicity | https://www.ncbi.nlm.nih.gov/geo/query/acc.cgi?acc=GSE255610 | NCBI Gene Expression Omnibus, GSE255610 |

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

# Appendix 1

## Experimental methods

### Identification and selection of gRNA

All gRNAs were designed following the same in silico identification algorithm. Genomic sequences were first identified using the UCSC Genome Browser and manipulated in Benchling. CRISPOR was used to separately identify all gRNA within a 500 bp window of each gene's TSS. TSS coordinates were determined using Phantom prediction, gene annotation, and DNase Hypersensitivity data as visualized in the Genome Browser with the hg38 genome assembly. gRNA were selected from this list to optimize for coverage, predicted specificity (CFD score ≥ 80), and predicted on-target activity (Doench 16' score) (*Doench et al., 2016*). When no gRNA were available in a region, predicted score constraints were minimally relaxed to identify a gRNA.

### Plasmid & guide cloning, transfection, mRNA extraction, and qPCR

dCas9-p300 was cloned into a lentiviral plasmid backbone (Addgene# 83889) was a gift from Gersbach lab. gRNA were cloned using the molecular cloning pipeline described by the Zhang group. These gRNA were cloned into an isogenic minimal guide expression backbone utilizing the Gecko guide cloning strategy (*Shalem et al., 2014*). This minimal guide cloning plasmid was a gift from Gersbach lab (Addgene#47108). Following sequence verification, gRNA tiling experiments were completed.

### Cell culture

HEK293T (ATCC, CRL-11268) were purchased from ATCC and cultured using supplemented DMEM (10% FBS [Millipore], 1% penicillin/streptomycin [Gibco]). These cells were initially expanded and cryopreserved in 0.5% (vol/vol) DMSO containing supplemented DMEM at a concentration of 2E6/ml per vial. HEK293T cells were consistently passaged at 80% confluence using Trypsin/EDTA (Gibco) dissociation and passaged at a 1:10 ratio. Cells were disregarded after their 10th passage. No Mycoplasma contamination was observed.

### gRNA tiling qPCR experiments

On day 0, healthy HEK293T cells (<passage 10) were lifted with Trypsin EDTA, centrifuged, resuspended, and counted with a manual hemocytometer using Trypan blue to assess health. Cells with >95% viability were seeded into 24-well plates with a consistent cell number per well ($1.5 \times 10^5$). 24 hours post plating, cells with confluence of between 70% and 90% with healthy phenotype were co-transfected with individual gRNA and dCas9-p300 plasmid DNA (mass = 500 ng, 125 ng, gRNA:375 ng dCas9-p300, using 1.5 ul Lipofectamine 3000) according to the manufacturer's protocol. All qPCR experiments were conducted using 24 samples (two biological samples per condition) measured in the 96-well format. Additionally, these experiments individually test 11 uniquely targeting gRNA and utilized a non-targeting gRNA as a negative control for downstream analysis. Total cell mRNA was extracted using the QIAGEN RNeasy kit and protocol. Reverse transcription was then carried out using iScript Advanced reverse transcriptase (Bio-Rad) 750 ng of total RNA in a 10 ul reaction. From there, cDNA was diluted to 10 ng/ul based on the initial total RNA input. Then qPCR reactions were assembled in technical duplicate and consisted of the following: 45 ng (original mass) of reverse transcribed and diluted cDNA, Luna qPCR Mastermix (NEB), forward primer, and reverse primer. The appropriate primer set was used to target (a) the gene intended for transcriptional modulation and (b) GAPDH, a ubiquitously expressed gene used to normalize input cDNA mass.

### MNase-seq

MNase sample processing was completed similarly to the previous methods (*Cui and Zhao, 2012*) with modifications. HEK293T cells were grown in parallel for >3 passages. Three biological replicates were processed together to minimize variance. Crosslinking was carried out on 20E7 HEK293T cells with 1% formaldehyde incubated for 10 minutes at 37°C prior to glycine quenching. Next, lysis and washing occurred, followed by nuclei isolation via 600 × *g* centrifugation. An initial optimization was performed on 2E6 purified nuclei using MNase amounts between 0.1 and 64 units of enzyme. RNase

treatment as well as Proteinase K treatment and removal of crosslinks were performed as previously described (*McKnight et al., 2021*). QIAquick PCR purification, sample DNA were visualized with the use of a 2% agarose gel and TapeStation (Agilent). The mononucleosomal band 150 bp was cut from the gel and purified using the QIAquick Gel purification. Heat was not used when melting gel to preserve AT rich regions. Following mono nucleosomal band purification, samples were quantified and size verified using a Tape Station (Agilent). Illumina libraries were produced using the NEBNext Ultra II DNA library preparation kit with NEBNext Dual Index Multiplex Oligos for Illumina using 1 µg of purified DNA as input and SPRI bead size selection after adapter ligation prior to index addition. Color-balanced unique i5 and i7 indices were used for each biological replicate to reduce confounds associated with index hopping. Prepared library concentrations and purity were determined on a tape station. Following verification, the three biological replicates were admixed with the same mass and sent to Azenta for sequencing on a single lane on the HISeq 3000/4000 platform with expected yield of 350 million paired end 150 bp length reads. This sequencing scheme was expected to yield a coverage of ~10× for each biological replicate sample.

## H3K27ac CUT&RUN qPCR

H3K27ac CUT&RUN qPCR CUT&RUN was completed using the CUTANA ChIC/CUT&RUN Kit by Epicypher (Catalog #: 14-1048). H3K27ac was bound using the Anti-Histone H3 (acetyl K27) antibody (Catalog#: ab4729) sold by Abcam. *Escherichia coli* spike-in DNA and Rabbit IgG (components of kit#: 14-1048) were used for qPCR input normalization and negative control, respectively. All experiments were performed in duplicate with three independent experiments. Briefly, p300 and individual gRNA were co-transfected into HEK293T cells, after 72 hours cells were detached, and CUT&RUN was completed with identical cell number were used for each sample. qPCR was completed in technical duplicate using a primer set designed for amplification near the targeted promoter (*CXCR4* or *TGFBR1*). qPCR reactions were also completed using a previously described primer set for quantification of the *E. coli* gene uida. The ddT relative qPCR methods were used to analyze data, where uida Ct was used to normalize input and fold over rabbit IgG was calculated for each sample.

## Perturb-seq scCRISPRa transduction

Data from Perturb-seq experiments were generated in a previous study (*Goell et al., 2024*). Briefly, monoclonal K562 cell lines were generated by transducing cells (8 ug/mL) with respective constructs of interest across a range of %v/v and performing flow cytometry 2 days later. Wells receiving dilutions with 40% mCherry+ were selected and plated for monoclonal lines by limiting dilution. Monoclonal lines were transduced with varying titers to assess viral copy number. Library transduction with a 1.5% v/v was selected for the scCRISPRa experiment in which 500k cells were transduced with virus containing the gRNA library. Cells were spun out of polybrene-containing media and resuspended in standard K562 culture media. At 2 days post-transduction, 1 ug/mL puromycin was added to the culture. 9 days after transduction, cells were collected for scRNA-seq.

## 10X Genomics scRNA-sequencing with gRNA capture

Cells were harvested and prepared as per the 10X Genomics Single Cell Protocols Cell Preparation Guide. 10,000 cells were captured per lane using a 10X Chromium device. One lane was used for dCas9-p300 WT containing cells. Cells were captured using a 10X Chromium chip using the Chromium Next GEM Single Cell 3' Reagents Kit v.3 with Feature Barcoding Technology for CRISPR screening. Final libraries were sequenced using a NovaSeq 6000 for each p300 screen. Gene expression and CRISPR Guide Capture transcript libraries were pooled at a 4:1 ratio for sequencing.

## Computational methods

### Analysis of Perturb-seq data

We analyzed single-cell RNA sequencing (scRNA-seq) data generated on the 10X Genomics platform as described previously (*Goell et al., 2024*), corresponding to $G$ genes and $r$ guide RNAs (gRNAs) across $C$ cells using the following steps:

## Quality control

- Cells were retained if the total gene count in a cell was between 1000 and 10,000, ensuring adequate complexity and excluding potential empty droplets or doublets.
- Cells with mitochondrial gene content exceeding 10% were excluded to avoid including dying or stressed cells.

## Normalization

To normalize the expression data, the following was applied to each gene's raw count in each cell:

$$\text{Normalized Expression} = \frac{\text{Raw Count}}{\text{Total Counts per Cell} + 1}$$

## Computing Perturb-seq gene expression

For each gene $g$, targeted by a set of gRNAs denoted as $r_g$, we determined the impact of each specific gRNA $r_g^i$ on its gene expression by performing the following steps:

- **Expression thresholding:** Only cells with non-zero expression of gene $g$ were selected for further analysis.
- **gRNA-specific selection:** From these cells, only those expressing the specific gRNA $r_g^i$ and none other from $r_g$ were retained.
- **Pseudobulk quantification:** For these cells, we computed the pseudobulk mean ($\mu$) and standard deviation ($\sigma$) of the expression levels of gene $g$.

## Computing Perturb-seq fold-change

To establish the baseline expression of gene $g$, we considered cells not targeted by any gRNAs $r_g^i$. The pseudobulk mean ($\mu_{\text{control cells}}$) and standard deviation ($\sigma_{\text{control cells}}$) of gene $g$'s expression in these cells were calculated. The fold-change for gene $g$ due to gRNA $r_g^i$ was then quantified as

$$\text{fold-change} = \frac{\mu}{\mu_{\text{control cells}}}$$

## Plotting Perturb-seq fold-change against model predictions

In order to prepare *Figure 6—figure supplement 4*, we performed the following steps:

- Genes were included if there were at least two distinct 25 bp bins within 250 base pairs of the TSS with a gRNA targeting that gene and having a distinct expression fold-change.
- gRNAs were included if the number of cells expressing $r_g^i$ was $\geq 2$.
- For each bin, the average Perturb-seq fold-change $\mu$ and the average predicted fold-change $\mu_{\text{predicted}}$ were calculated as:

$$\mu_{\text{bin}} = \frac{\sum \text{fold-change}}{\text{number of fold-change observations in the bin across gRNAs targeting the same 25 bp bin}}$$

$$\mu_{\text{predicted, bin}} = \frac{\sum \text{predicted fold-change}}{\text{number of models used to make a prediction for the fold-change within this bin}}$$

- Ranks were assigned to $\mu_{\text{bin}}$ and $\mu_{\text{predicted, bin}}$ for comparison.
- These ranks were then plotted against each other to evaluate the correlation between observed Perturb-seq fold-change and the model-predicted fold-change.

**Appendix 1—table 1.** ChIP-seq $-\log_{10}(\text{p-values})$ were obtained from the ENCODE Imputation Challenge where the ground truth data were available (corresponding to entries labeled T in the table).

Avocado imputations were downloaded from the ENCODE data portal, where ground truth data were not available (corresponding to entries labeled A in the table).

| Cell type | polyA Plus RNA-seq | H3K36me3 | H3K27me3 | H3K27ac | H3K4me1 | H3K4me3 | H3K9me3 |
|---|---|---|---|---|---|---|---|
| IMR-90 | T | T | T | T | T | T | T |

*Appendix 1—table 1 Continued on next page*

*Appendix 1—table 1 Continued*

| Cell type | polyA Plus RNA-seq | H3K36me3 | H3K27me3 | H3K27ac | H3K4me1 | H3K4me3 | H3K9me3 |
|---|---|---|---|---|---|---|---|
| H1-hESC | T | T | T | T | T | T | T |
| Trophoblast cell | T | T | T | T | T | T | T |
| Neural stem progenitor cell | T | T | T | T | T | T | T |
| K562 | T | T | T | T | T | T | T |
| Heart left ventricle | T | T | T | T | T | T | T |
| Adrenal gland | T | T | T | T | T | T | T |
| Endocrine pancreas | T | T | T | T | T | T | T |
| Peripheral blood mononuclear cell | T | T | T | T | T | T | T |
| Amnion | T | T | T | T | T | T | T |
| Myoepithelial cell of mammary gland | T | T | T | A | T | T | T |
| Chorion | T | T | T | T | T | T | A |
| HEK293 | T | T | A | T | T | T | T |

**Appendix 1—table 2.** Endogenous gene expression of genes for which we generated dCas9-p300 epigenome editing data indicates that genes for which high fold-change was obtained are more likely to have low endogenous gene expression in HEK293T. Cross-cell type Spearman provides a metric to assess how accurate our CNN model predictions are, on any given gene, across the 13 cell types.

| Gene | HEK293 (SRR3997504) TPM | HEK293T (SRR13341848) TPM | HEK293T (SRR15013784) TPM | Maximum fold-change in dCas9-p300 data | Cross-cell type Spearman |
|---|---|---|---|---|---|
| PRSS12 | 12.710 | 8.448 | 6.910 | 2.380 | 0.896 |
| CXCR4 | 11.974 | 2.826 | 8.216 | 5.365 | 0.852 |
| TGFBR1 | 0.725 | 3.254 | 8.029 | 3.675 | 0.689 |
| C2CD4B | 0.306 | 0.000 | 0.000 | 591.312 | 0.726 |
| CD79A | 0.280 | 0.207 | 0.127 | 127.094 | 0.364 |
| SOX11 | 0.051 | 0.131 | 0.209 | 14.245 | 0.846 |
| MYO1G | 0.000 | 0.016 | 0.000 | 37.948 | 0.621 |
| CYP17A1 | 0.000 | 0.000 | 0.000 | 6,549.110 | 0.397 |

