## [Editor Report · eLife Assessment]

This study presents an advance in efforts to use histone post-translational modification (PTM) data to model gene expression and predict epigenetic editing activity. Such models are broadly **useful** to the research community, especially ones that can model and predict epigenetic editing activity, which is novel; additionally, the authors have nicely integrated datasets across cell types into their model. The work is mostly **solid**, but it would be strengthened by performing further comparisons to existing methods that predict gene expression from PTM data and from more comprehensive functional validation of model-predicted epigenome editing outcomes beyond dCas9-p300 based perturbations. This work will be of interest to the epigenetics and computational modeling communities.

---

## [Referee Report · Reviewer #1 (Public review)]

Batra, Cabrera and Spence et al. present a model which integrates histone posttranslational modification (PTM) data across cell models to predict gene expression with the goal of using this model to better understand epigenetic editing. This gene expression prediction model approach is useful if (a) it predicts gene expression in specific cell lines (b) it predicts expression values rather than a rank or bin, (c) if it helps us to better understand the biology of gene expression or (d) it helps us to understand epigenome editing activity. Problematically for points (a) and (b) it is easier to directly measure gene expression than to measure multiple PTMs and so the real usefulness of this approach mostly relates to (c) and (d).

Other approaches have been published that use histone PTM to predict expression (e.g. PMID 27587684, 36588793). Is this model better in some way? No comparisons are made, although a claim is made that direct comparisons are difficult. I appreciate that the authors have not used the histone PTM data to predict gene expression levels of an "average cell" but rather that they are predicting expression within specific cell types or for unseen cell types. Approaches that predict expression levels are much more useful, whereas some previous approaches have only predicted expressed or not expressed or a rank order or bin-based ranking. The paper does not seem to have substantial novel insights into understanding the biology of gene expression.

The approach of using this model to predict epigenetic editor activity on transcription is interesting and to my knowledge novel although only examined in the context of a p300 editor. As the author point out the interpretation of the epigenetic editing data is convoluted by things like sgRNA activity scoring and to fully understand the results likely would require histone PTM profiling and maybe dCas9 ChIP-seq for each sgRNA which would be a substantial amount of work.

Furthermore from the model evaluation of H3K9me3 is seems the model is performing modestly for other forms of epigenetic or transcriptional editing- e.g. we know for the best studied transcriptional editor which is CRISPRi (dCas9-KRAB) that recruitment to a locus is associated with robust gene repression across the genome and is associated with H3K9me3 deposition by recruitment of KAP1/HP1/SETDB1 (PMID: 35688146, 31980609, 27980086, 26501517).

One concern overall with this approach is that dCas9-p300 has been observed to induce sgRNA independent off target H3K27Ac (https://www.ncbi.nlm.nih.gov/pmc/articles/PMC8349887/ see Figure S5D) which could convolute interpretation of this type of experiment for the model.

Comments on revisions: This resubmission adds a comparison to existing gene prediction methods, but add no new confirmation experiments with predicting epigenome editing efficiency and had only one minor text edit.

---

## [Referee Report · Reviewer #2 (Public review)]

Summary:

The authors build a gene expression model based on histone post-translational modifications, and find that H3K27ac is correlated with gene expression. They compare to other gene prediction methods such as DeepChrome. They proceed to perturb H3K27ac at 13 gene promoters in two cell types, and measure gene expression changes to test their model.

Strengths:

The combination of multiple methods to model expression, along with utilizing 6 histone datasets in 13 cell types allowed the authors to build a model that correlates between 0.7-0.79 with gene expression.

They compare three cells types to other prediction models, and this figure should be included in the main figures.

They use dCas9-p300 fusions to perturb H3K27ac and monitor gene expression to test their model. Ranked correlations of the HEK293 data showed some support for the predictions after perturbation of H3K27ac.

Weaknesses:

The authors state in the latest submission that the primary use case of this work is related to predicting epigenome editing outcomes, not predicting gene expression from chromatin. However the first four figures all relate to gene expression prediction. The only main figure that shows epigenome editing prediction is panel 6E. If this authors wish to highlight the use case of this work they should redo figures, including moving panels from current supplemental figures to show this.

The perturbation of 5 genes in K562 with perturb-seq data shows a modest correlation of ~0.5 and is still only shown in supplemental figures, which is odd as this is the true test case of their model in my opinion. The authors are then left to speculate the reasons why the outcome of epigenome editing doesn't fit their predictions, which highlights the limited value in the current version of this method.

As mentioned before, testing genes that were not expressed being most activated by dCas9-p300 weaken the correlations vs. looking at a broad range of different gene expression as the original model was trained on.

If the authors want this method to be used to predict outcomes of epigenome editing, expanding to dCas9-KRAB and other CRISPRa methods (SAM and VPR) would be useful. Those datasets are published and could be analyzed for this manuscript and show how the model holds up across cell types and epigenome editing methods.

The utility of this method as described here, to predict gRNA outcomes seems modest and limited. It is fairly trivial to test 10 or more gRNAs for a single gene to find the best one, and the authors show limited prediction and occasionally no benefit. For example, with CHD8 and CD79 the gRNA with the highest prediction had the lowest actual impact on gene expression of the gRNAs tested. For many other genes the gRNA's prediction and gene expression outcome show no correlation.

---

## [Author Response]

The following is the authors’ response to the previous reviews

**Reviewer #1 (Public Review):**
Batra, Cabrera and Spence et al. present a model which integrates histone posttranslational modification (PTM) data across cell models to predict gene expression with the goal of using this model to better understand epigenetic editing. This gene expression prediction model approach is useful if (a) it predicts gene expression in specific cell lines (b) it predicts expression values rather than a rank or bin, (c) if it helps us to better understand the biology of gene expression or (d) it helps us to understand epigenome editing activity. Problematically for point (a) and (b) it is easier to directly measure gene expression than to measure multiple PTMs and so the real usefulness of this approach mostly relates to (c) and (d).

We appreciate this point from Reviewer #1 and the instructive comments and helpful feedback on our study. We designed our approach keeping in mind that the primary use case is to understand how epigenome editing would affect gene expression.

Other approaches have been published that use histone PTM to predict expression (e.g. PMID 27587684, 36588793). Is this model better in some way? No comparisons are made although a claim is made that direct comparisons are difficult. I appreciate that the authors have not used the histone PTM data to predict gene expression levels of an "average cell" but rather that they are predicting expression within specific cell types or for unseen cell types. Approaches that predict expression levels are much more useful whereas some previous approaches have only predicted expressed or not expressed or a rank order or bin-based ranking. The paper does not seem to have substantial novel insights into understanding the biology of gene expression.

We thank Reviewer #1 again for this insightful comment. We have included citations for a series of papers (PMIDs: 27587684, 30147283, 36588793) that performed gene expression prediction using histone PTM data. However, each of these methods performs classification of gene expression as opposed to predicting the actual gene expression value via regression. Additionally, the referenced studies all work with Roadmap Epigenomics read-depth data as opposed to p-values obtained from the ENCODE pipelines, making it difficult to make direct comparisons. We outline in the Discussion section that by creating a comprehensive dataset of epigenome editing outcomes, which include quantification of histone PTMs before and after in situ 1 perturbations, will improve our understanding of the effects of dCas9-p300 on gene expression and assist in the design of gRNAs for achieving fine-tuned control over gene expression levels. In this revised version of our study, we have also added new data (Figure 3 – figure supplement 3) to further benchmark our model against others.

The approach of using this model to predict epigenetic editor activity on transcription is interesting and to my knowledge novel although only examined in the context of a p300 editor. As the author point out the interpretation of the epigenetic editing data is convoluted by things like sgRNA activity scoring and to fully understand the results likely would require histone PTM profiling and maybe dCas9 ChIP-seq for each sgRNA which would be a substantial amount of work.

We agree with the Reviewer and view these experiments as important components of future studies.

Furthermore from the model evaluation of H3K9me3 is seems the model is performing modestly for other forms of epigenetic or transcriptional editing- e.g. we know for the best studied transcriptional editor which is CRISPRi (dCas9-KRAB) that recruitment to a locus is associated with robust gene repression across the genome and is associated with H3K9me3 deposition by recruitment of KAP1/HP1/SETDB1 (PMID: 35688146, 31980609, 27980086, 26501517).

This is an interesting point. We have included new data (Figure 4 – figure supplement 1), that quantifies how sensitive the trained gene expression model is to perturbations in H3K9me3. Indeed our data suggests that the model predictions are sensitive to perturbations in H3K9me3. For instance, there is a clear decrease and a gradual increase as the position where the perturbation is performed moves from upstream to downstream of the TSS. Additionally, the magnitude of the predicted fold-change is a function of how much the H3K9me3 is perturbed and hence the magnitude of change would be even higher if the perturbation magnitude is increased. However, this precise magnitude is hard to estimate In the absence of experimental perturbation data for H3K9me3. Leveraging our model in combination with KRAB-based CRISPRi is an exciting and important aspect of future studies.

One concern overall with this approach is that dCas9-p300 has been observed to induce sgRNA independent off target H3K27Ac (https://www.ncbi.nlm.nih.gov/pmc/articles/PMC8349887/ see Figure S5D) which could convolute interpretation of this type of experiment for the model.

This remains an excellent point and indeed, we and others have observed that dCas9-p300 can result in off-target H3K27ac levels (both increased and suppressed) across the genome. Our study focused on p300, because the molecule is one of the few known proteins that can catalyze H3K27ac in the human genome, and H3K27ac remains a proxy for active genomic regulatory elements. Nevertheless, any off target activity of dCas9-p300 could certainly convolute our analyses. We have included language to address this caveat in our discussion.

**Reviewer #2 (Public review):**
Summary:The authors build a gene expression model based on histone post-translational modifications, and find that H3K27ac is correlated with gene expression. They proceed to perturb H3K27ac at 13 gene promoters in two cell types, and measure gene expression changes to test their model.

We remain appreciative of the constructive feedback and input from Reviewer #2 on our manuscript.

Strengths:The combination of multiple methods to model expression, along with utilizing 6 histone datasets in 13 cell types allowed the authors to build a model that correlates between 0.7-0.79 with gene expression. They use dCas9-p300 fusions to perturb H3K27ac and monitor gene expression to test their model. Ranked correlations of the HEK293 data showed some support for the predictions after perturbation of H3K27ac.Weaknesses:The perturbation of 5 genes in K562 with perturb-seq data shows a modest correlation of ~0.5 and isn't included in the main figures. The authors are then left to speculate reasons why the outcome of epigenome editing doesn't fit their predictions, which highlights the limited value in the current version of this method.

We agree with the reviewer’s suggestion and highlight in our conclusion that generating epigenome editing data across a variety of cell types and across many genes will help uncover the underlying mechanisms of gene expression modulation.

As mentioned before, testing genes that were not expressed being most activated by dCas9-p300 weaken the correlations vs. looking at a broad range of different gene expression as the original model was trained on.

We appreciate this comment from Reviewer #2. We note that the data generated from this dCas9-p300 perturb-seq experiment used gRNAs from a pre-existing library published previously (PMID: 37034704). While this library enabled deeper interrogation of dCas9-p300 driven effects compared to our previous revision, the gRNAs in this library were designed against genes associated with haploinsufficiency in neuronal cell types, and which were generally lowly-expressed in K562 cells. Further, we restricted our analysis here to promoter-proximal gRNAs (as opposed to enhancer-targeted gRNAs in the library), focusing our scope even more so. Thus the genes ultimately used for analysis are enriched for low expression.

If the authors want this method to be used to predict outcomes of epigenome editing, expanding to dCas9-KRAB and other CRISPRa methods (SAM and VPR) would be useful. Those datasets are published and could be analyzed for this manuscript.

This is an exciting suggestion from Reviewer #2. We agree, and view this as a component of future work in this area.

The authors don't compare their method to other prediction methods.

In this revised version of our study, we have also added new data (Figure 3 – figure supplement 3) to further benchmark our model against others. These data demonstrate that our CNN model outperforms existing approaches across multiple cell types.

**Recommendations for the authors:**

**Reviewer #2 (Recommendations for the authors):**
Looking at the individual genes in K562 shows a random looking range of predictions and observed, with the exception of Bcl11A which is one of two genes in this set of 5 that are not expressed. I will repeat my earlier comment, that epigenome editing and CRISPRa methods generally show the most upregulation with the lowest expressed genes. I speculate that plotting endogenous expression vs. outcome (assuming using all gRNAs within a reasonable and similar distance to TSS) would produce a correlation of -0.5 or greater and be as useful as this method.

We agree, and believe that this demonstrates more work is needed in this emerging research area.

The methods describe Perturb-seq analysis but not the bench experiments.

We have added the bench methods related to our Perturb-seq experiments to our revised manuscript under the Experimental Methods section in the Appendix.

I don't understand why the authors can't compare to other methods as that is fairly standard in new prediction papers. I get that others used REMC vs. ENCODE, and were rank or binary based, but the authors could use REMC data and/or convert their data to ranked or binary and still compare. Lacking that it's hard to judge this manuscript.

We have added benchmarking against existing methods as Figure 3 – figure supplement 3.